# ADReLU: Enhancing Neural Network Expressivity with Attention-based Dynamic ReLU

## Abstract

Deep Neural Networks (DNNs) rely on activation functions to introduce non-linearity, which significantly impacts performance across various tasks. To enhance neural network expressivity, we propose **Attention-based Dynamic ReLU (ADReLU)**—a novel activation function that replaces ReLU's fixed zero threshold with a dynamic, input-dependent threshold computed via an attention mechanism. To balance expressivity and computational efficiency, ADReLU employs grouped convolution and depth-wise projection for image data, reduce the computational cost typically associated with attention. Extensive experiments on CIFAR-10, CIFAR-100, SVHN, and ImageNet datasets demonstrate that ADReLU consistently outperforms both predefined activation functions (such as ReLU, LReLU) and trainable (such as PReLU, GCLU, GELU, Maxout, and Dynamic ReLU) in terms of accuracy. Furthermore, we empirically analyze ADReLU's attention subspace dimension, sparsity patterns, and computational complexity, highlighting its balanced efficacy in feature representation and resource efficiency.

## 1 Introduction

Deep Neural Networks (DNNs) have demonstrated substantial success across various real-world tasks, including image classification He et al. (2024), natural language modeling Zhang et al. (2025), and computer vision Zhou et al. (2024). A fundamental component of DNNs is the activation function, which introduces non-linearity and enables the network to learn complex patterns. Hence, the choice of activation function plays a crucial role in the performance of DNNs, as it directly influences the network's capacity to model intricate relationships in the data Zhao et al. (2024). Here, we explore the potential of enhancing neural network expressivity via a novel activation function.

Activation functions, integral to neural networks, are broadly classified into predefined and trainable categories. Predefined activation functions, such as the Rectified Linear Unit (ReLU) Nair & Hinton (2010), Leaky ReLU (LReLU) Maas et al. (2013), rely on fixed parameters. In contrast, the trainable activation functions incorporate learnable parameters to increase the expressivity and flexibility. Trainable activation functions can be further divided into input-independent functions, such as Parametric ReLU (PReLU) He et al. (2015), which learn a static slope per neuron, and input-dependent functions, such as Dynamic ReLU (Dy-ReLU) Chen et al. (2020), which adjust parameters based on input data during both training and inference.

Among the proposed trainable activation functions in previous studies, there are some limitations. For example, input-independent functions like PReLU remain fixed during inference, restricting their ability to adapt to diverse input distributions. Input-dependent functions, such as Funnel ReLU (FReLU) Qiu et al. (2018) and Dynamic ReLU (Dy-ReLU) Chen et al. (2020), offer greater adaptability but typically compute a single parameter applied uniformly across channels or spatial locations. This uniform parameterization limits their capacity to capture fine-grained, channel-specific, or spatially varying features that are critical for complex data distributions in image classification tasks.

To address these limitations, we propose a novel activation function called **Attention-based Dynamic ReLU (ADReLU)**. The core idea behind ADReLU is to replace ReLU's static zero threshold with a dynamic, input-dependent threshold, computed via an attention mechanism. Unlike existing trainable activation functions, ADReLU computes a unique, input-dependent threshold $\tau$ for each

element via a QKV-inspired attention mechanism Vaswani et al. (2017), enabling the network to capture global channel-wise and local spatial interactions within the data. To reduce the computational cost of attention operations, the implementation of ADReLU integrates grouped convolution and depthwise projections, while maintaining accuracy.

In summary, the main contributions are as follows:

1. To enhance the expressivity of neural network models, we propose ADReLU, an input-dependent activation function that leverages an attention mechanism to replace ReLU's fixed zero threshold with dynamically computed thresholds.

2. To reduce the overhead of computing our ADReLU, we present an efficient implementation of ADReLU via using grouped convolution and depth-wise projection.

We have conducted a comprehensive empirical evaluation of our ADReLU, demonstrating its effectiveness across lightweight and deep architectures on diverse datasets, while analyzing sparsity patterns to optimize feature representation.

## 2  RELATED WORK

Activation functions are commonly non-linear, essential for enabling deep neural networks (DNNs) to model complex patterns, and broadly categorized as predefined and trainable.

**Predefined Activation Function**. To introduce non-linearity, predefined activation functions use fixed parameters. For instance, the Rectified Linear Unit (ReLU), defined as $\text{ReLU}(x) = \max(x, 0)$, is widely adopted due to its simplicity, computational efficiency, and ability to mitigate vanishing gradient issues Nair & Hinton (2010). However, ReLU's fixed zero threshold can lead to the "dying ReLU" problem Maas et al. (2013), where neurons become inactive for negative inputs, reducing the model's expressiveness. Variants like LReLU Maas et al. (2013) address this by introducing a small constant slope $\alpha$ for negative inputs, defined as $\text{LReLU}(x) = \max(x, \alpha x)$, allowing non-zero gradients for negative inputs. Despite this improvement, LReLU's constant slope requires manual tuning, which may be suboptimal across diverse tasks or datasets.

**Trainable Activation Function**. The trainable activation functions incorporate learnable parameters to enhance flexibility and can be further divided into input-independent and input-dependent approaches. Input-independent trainable activation functions learn static parameters that remain fixed during inference. Such as PReLU He et al. (2015), generalizes LReLU by learning a slope $\alpha$ for each neuron. Flexible ReLU Qiu et al. (2018) introduces a learnable bias term $b$ to adjust the activation's shape. Maxout Goodfellow et al. (2013) generalizes ReLU by dividing the input into groups and outputting the maximum value. And GCLU Xu et al. (2025) leverages Gaussian distributions to calibrate input responses. Although these methods outperform predefined activation functions upon their reported results, their static parameters limit their adaptability to varying input distributions during inference.

In contrast, input-dependent activation functions dynamically adjust their parameters based on input data samples. For example, Funnel ReLU (FReLU) (Ma et al., 2020) computes piecewise linear parameters using a spatial condition. While the Dy-ReLU Chen et al. (2020) employs a hyper-function to generate parameters for a piecewise linear function, and achieves significant performance improvements but introduces complexity due to the use of two fully connected layers as a hyper-function to process input features. These methods improve model performance by tailoring activation behavior to input data. However, they typically apply a single parameter uniformly across channels or spatial locations, lacking mechanisms to capture fine-grained, channel-specific, or spatially varying patterns.

## 3  ATTENTION-BASED DYNAMIC RELU

The Attention-based Dynamic ReLU (ADReLU) is a novel input-dependent activation function, as illustrated by Fig. 1 (a), the core principle of ADReLU is an element-wise maximum operation that chooses between the input $x$ and a dynamic, input-dependent threshold $\tau$. Unlike ReLU, which applies a fixed threshold of zero, ADReLU adapts its threshold $\tau$ based on the input features, enabling more flexible and context-aware non-linear transformations.

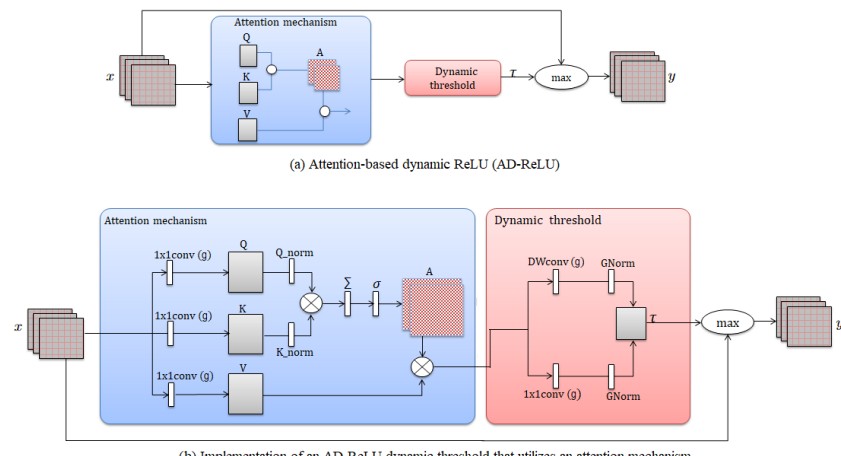

Figure 1: Attention-based Dynamic ReLU (ADReLU); (a) the core principle of ADReLU; (b) the implementation of an ADReLU for the image classification task.

Formally, given an input tensor $\boldsymbol{x}$, ADReLU produces the output tensor $\boldsymbol{y}$ of identical dimension through an element-wise operation, shown by Eq.(1):

$$\boldsymbol{y} = \max\{\boldsymbol{x}, \boldsymbol{\tau}\}, \tag{1}$$

where $\boldsymbol{\tau} = f(\boldsymbol{x}; \Theta)$ is a dynamic threshold tensor of identical dimensions generated by the function $f$ parameterized by $\Theta$, and the function $f$ exploits the attention mechanism to enable input-dependent adaptation.

**Implementation of ADReLU or the image classification task**. For different tasks, we implement ADReLU based on an attention mechanism inspired by the $QKV$ framework in 2 steps, shown by Fig. 1 (b).

Firstly, the input tensor $\boldsymbol{x}$ is converted into the attention scores $A$ by Eq. (2),

$$(Q, K, V) = QKV_{conv}^{(g)}(\boldsymbol{x}), \quad Q = \frac{Q}{\|Q\|_2 + \epsilon}, \quad K = \frac{K}{\|K\|_2 + \epsilon}, \quad A = \sigma(\sum_{i=1}^{d_k} Q_i \cdot K_i)V, \tag{2}$$

where the input tensor $\boldsymbol{x}$ is projected into Query $(Q)$, Key $(K)$, and Value $(V)$ tensors using a grouped $1 \times 1$ convolution, denoted as $QKV_{conv}^{(g)}$, $g$ is the group size which indicates all channels are split into $g$ groups, each processing as an attention subspace of dimension $d_k$; the $\| \cdot \|_2$ is L2 normalization, denoted by $Q_{\text{norm}}$, $K_{\text{norm}}$ in Figure 1 (b), $\epsilon$ is a small constant value to promote numerical stability and prevent exploding gradients; the attention scores $A$ are computed via element-wise multiplication and summation along the $d_k$ dimension, followed by a sigmoid function $\sigma(\cdot)$.

Secondly, the dynamic threshold $\boldsymbol{\tau}$ is generated by processing the attention scores $A$ through a depthwise separable projection, as shown by Eq. (3-6),

$$Z_{dw} = \text{DW}_{conv}^{(g)}(A), \tag{3}$$

$$\hat{Z}_{dw} = \text{GNorm}^{(g_{dw})}(Z_{dw}), \tag{4}$$

$$T = \text{conv}_{1 \times 1}^{(g)}(\hat{Z}_{dw}), \tag{5}$$

$$\boldsymbol{\tau} = \text{GNorm}^{(g)}(T), \tag{6}$$

where $\text{DW}_{conv}^{(g)}$ denotes a depthwise convolution with a kernel size of $3 \times 3$ to capture local spatial patterns, followed by group normalization $\text{GNorm}^{(g_{dw})}$ for stability across groups. A final $\text{conv}_{1 \times 1}^{(g)}$ projects the features, and another group normalization $\text{GNorm}^{(g)}$ is applied to produce $\tau$.

This depthwise projection allows $\boldsymbol{\tau}$ to adapt not only to the global input content but also to local spatial and channel-specific patterns, significantly enhancing the network's expressivity while maintaining efficiency.

*Remarks*. Here, we present more detailed implementations based on CNNs and Vision Transformers by example, respectively. To reduce the time cost of computing the standard self-attention mechanism, which is quadratic computational complexity $\mathcal{O}((HW)^2 C)$ with respect to spatial dimensions, we explain our implementations with linear complexity $\mathcal{O}(CHW)$.

In case of CNNs, the input tensors $\boldsymbol{x} \in \mathbb{R}^{B \times C \times H \times W}$. ADReLU first uses a $\text{conv}_{1 \times 1}^{(g)}$ to project the input into separate $Q, K, V$ tensors. The subsequent attention mechanism, which operates on these $Q, K, V$ projections, captures both local spatial and global channel-wise feature interactions, overcoming the limitations of static activations. The complexity of these operations scales as $\mathcal{O}(CHW)$ for $\text{conv}_{1 \times 1}^{(g)}$, and the depthwise convolutions $\text{DW}_{conv}^{(d_k)}$ scaling as $\mathcal{O}(d_k CHW)$. And the final element-wise maximum has a complexity of $\mathcal{O}(CHW)$, making ADReLU significantly more efficient than traditional convolutions, which have a complexity $\mathcal{O}(C^2 HW)$.

For Vision Transformers (ViT-Base), the input is represented as a sequence of patches $\boldsymbol{x} \in \mathbb{R}^{B \times N \times C}$, where $N$ denotes the number of patches and $C$ is the dimensionality of each patch. The ViT inherently treats these patches as spatial units, analogous to pixels in a CNN feature map. ADReLU operates directly on this structure by considering the patch dimension $N$ as its spatial axis. The computational complexity of ADReLU in this context is $\mathcal{O}(d_k CN)$, where $N = (H \cdot W)/P^2$ for an original image size $H \times W$ and patch size $P$. As in CNNs, the cost is dominated by the projection and depthwise operations, scaling with the sequence length $N$ rather than spatial dimensions $H$ and $W$ directly. This design preserves computational efficiency, dynamic threshold adaptation, and activation sparsity, making ADReLU suitable for vision transformers.

## 4 EXPERIMENTAL STUDIES

To validate the effectiveness of our ADReLU, we replace the activation functions in the SOTA and popular neural networks, such as VGG, ResNet, SENet, MobileNet, ViT, Swin, and CaiT, with our ADReLU, and evaluate on image classification datasets CIFAR-10, CIFAR-100, SVHN, and ImageNet by comparison studies. Additionally, we assess the computational complexity of ADReLU relative to baseline activation functions as ReLU and Dy-ReLU, across ResNet variants. Furthermore, to investigate the underlying mechanisms of ADReLU, we examine the efficacy of ADReLU in lightweight fully connected architectures through case studies and analyze sparsity patterns that effectively balance expressivity and regularization.

### 4.1 SETTINGS

**Datasets**. We evaluate the performance of the proposed Attention-based Dynamic ReLU (ADReLU) through comprehensive experiments on four benchmark image classification datasets: CIFAR-10, CIFAR-100 Krizhevsky et al. (2009), Street View House Numbers (SVHN) Netzer et al. (2011), and ImageNet Deng et al. (2009); Russakovsky et al. (2015). **CIFAR-10 and CIFAR-100** each consist 60,000 RGB images of size $(32 \times 32)$, with 50,000 images for training and 10,000 for testing. CIFAR-10 contains 10 distinct classes, whereas CIFAR-100 includes 100 classes, making it more challenging. The **SVHN** dataset consists of real-world digit images from Google Street View with 630,420 RGB images $32 \times 32$ pixels of digits from 0 to 9, split into 73,257 training images, 26,032 test images, and an additional 531,131 images available for extended training. **ImageNet** Deng et al. (2009); Russakovsky et al. (2015), specifically the ILSVRC 2012 subset, includes 1,281,167 training images and 50,000 validation images distributed across 1,000 distinct classes.

**Baselines**. Our experiments assess ADReLU effectiveness, computational efficiency, and adaptability compared to baseline activation functions, including predefined functions (ReLU, LeakyReLU) and trainable functions (PReLU, GELU, GCLU, Maxout, Dy-ReLU ). For different datasets, the SOTA models are different, we use a diverse set of neural network architectures, including VGG-16 Simonyan & Zisserman (2014), ResNet He et al. (2016), SENet-32 Hu et al. (2018), MobileNetV2 (MNetv2)Sandler et al. (2018); Howard et al. (2017), Vision Transformer (ViT) Dosovitskiy et al. (2020), Swin Transformer Liu et al. (2021), and Cait Touvron et al. (2021).

**Parameters**. By replacing their standard activation functions with our ADRelu, all models are trained using stochastic gradient descent (SGD) with a momentum of 0.9. Training spans 200 epochs with a batch size of 128 across CIFAR-10, CIFAR-100, and SVHN datasets, incorporating a 5-epoch

warm-up period to stabilize initial training. For CNN-based models, the initial learning rate is set to 0.1 for CIFAR-10, CIFAR-100, and SVHN, while ViT-Base uses 0.001 on CIFAR-10 and CIFAR-100 to suit its convergence properties. A cosine annealing learning rate scheduler Loshchilov & Hutter (2016) dynamically adjusts the learning rate. For MobileNetV2 (MNetv2) on ImageNet, the initial learning rate is 0.05, reduced to zero over a single cosine cycle, with training extended to 300 epochs, label smoothing of 0.1, weight decay of 2e-5, and dropout of 0.1 for width $\times 0.35$. For width $\times 0.5$, weight decay increases to 3e-5 and dropout to 0.2. Random cropping, flipping, and color jittering are applied for all width multipliers. For ResNet on ImageNet, the initial learning rate is 0.1, reduced by a factor of 10 at epochs 30 and 60, with a weight decay of 1e-4 and training for 90 epochs. A dropout rate of 0.1 is applied before the final layer, with label smoothing for ResNet-18, ResNet-34, and ResNet-50. For ADReLU, the group size is set to $g = 4$, and the attention subspace dimension is $d_k = 8$, optimized through preliminary experiments for performance and computational efficiency. All models are implemented in PyTorch.

## 4.2 COMPARISON STUDIES

To present comprehensive empirical comparisons of ADReLU against predefined and trainable activation functions, we compare their top-1 accuracies on the typical image classification tasks via integrating our ADReLU into the popular and SOTA neural networks, which are grouped into two types: the models trained from scratch and the pretrained models.

***ADReLU evaluations on the models trained from scratch***

Firstly, we evaluate the performance of ADReLU on four image classification benchmarks: CIFAR-10, CIFAR-100, SVHN, and ImageNet, using state-of-the-art (SOTA) models trained from scratch. To ensure fair and consistent comparisons, we report previously published accuracy scores for certain baseline activation functions. Values marked with an asterisk (*) in Tables 1, 2, and 3 are taken from Xu et al. (2025). Similarly, the ImageNet baseline results for ReLU and Dy-ReLU in Table 4 are sourced from Xu et al. (2025) and Chen et al. (2020), with the same training setups as ours, respectively. All other accuracy values (those without an asterisk) are obtained from our own experiments, where we reimplemented the activation functions to ensure consistency. The backbone model implementations were taken from widely used open-source repositories, which we provide at `https://github.com/tcmyxc/CV-Tutorial`. We trained these models using the same training pipeline as our ADReLU variants to maintain fairness.

The tables are organized with rows representing the SOTA models and columns representing the activation functions. Each cell contains the top-1 test accuracy (%) for the corresponding architecture–activation function pair, with the bolded values indicating the best performance.

**SVHN Dataset**. The main objective of this experiment is to evaluate ADReLU's adaptability to natural noise, varying lighting, diverse backgrounds that introduce variability and distribution shifts characteristic of the SVHN dataset.

Table 1: Top-1 accuracy (%) on the SVHN dataset for various activation functions across multiple network models

| Networks | ReLU | LReLU | PReLU | GELU | GCLU | Maxout | Dy-ReLU | ADReLU |
|----------|------|-------|-------|------|------|--------|---------|--------|
| VGG-16 | 95.94 | 95.83 | 95.64 | 95.59 | 95.62 | 96.32 | 95.81 | **96.78** |
| ResNet-8 | 93.97 | 93.67 | 93.59 | 93.50 | 94.39 | 94.79 | 95.01 | **95.55** |
| ResNet-32 | 95.82 | 96.47 | 96.48 | 96.61 | 96.68 | 96.48 | 96.45 | **96.99** |
| ResNet-50 | 96.29* | 96.03 | 96.84 | 96.22 | 96.62 | 95.96 | 96.45 | **96.98** |
| SENet-32 | 95.77* | 95.43 | 96.12 | 96.32 | 96.13 | 96.39 | 96.31 | **96.83** |
| MNetv2 | 95.76 | 95.68 | 95.65 | 95.83 | 95.54 | 95.92 | 95.56 | **96.31** |
| ViT-Tiny | 92.81 | 92.16 | 91.76 | 91.94 | 92.86 | 91.41 | 92.02 | **94.89** |
| Swin-Tiny | 90.61 | 89.70 | 90.29 | 89.88 | 90.35 | 88.79 | 89.27 | **91.51** |
| Cait-XXS | 91.50 | 89.87 | 90.05 | 91.36 | 89.87 | 89.64 | 90.27 | **94.18** |

As shown in Table 1, VGG-16 attains an accuracy of 96.78%, outperforming Maxout, which scored 96.32%, by 0.46%. On ResNet-8, ADReLU reaches 95.55%, surpassing Dy-ReLU's 95.01% by a margin of 0.54%. The trend continues with deeper architectures: ResNet-32 achieves 96.99%, exceeding GCLU's result of 96.68% by 0.31%. Similarly, ResNet-50 and SENet-32 with ADReLU

achieve 96.98% and 96.83%, outperforming Dy-ReLU by 0.53% and 0.52%, respectively. Furthermore, MobileNetV2 with a scale of 0.35, referred to in table MNetv2, has 96.31%, a 0.48% gain over GELU. Most notably, it delivers substantial improvements for vision transformers, boosting ViT-Tiny to 94.89% (+2.03%), Swin-Tiny to 91.51% (+1.16%), and CaiT-XXS to 94.18% (+2.82%) over their best respective baselines.

These results indicate that ADReLU adapts on an element-wise basis to capture channel and spatial-specific interactions within noisy data, thereby enabling reliable digit recognition in unstructured, real-world environments.

**CIFAR-10 and CIFAR-100**. The experiments on CIFAR-10 (coarse-grained) and CIFAR-100 (fine-grained) evaluate ADReLU's ability to enhance feature learning across different class granularities and architectures.

Table 2: Top-1 accuracy (%) on the CIFAR-10 dataset for various activation functions across multiple network models.

| Networks | ReLU | LReLU | PReLU | GELU | GCLU | Maxout | Dy-ReLU | ADReLU |
|---|---|---|---|---|---|---|---|---|
| VGG-16 | 93.86* | 93.80 | 93.06 | 93.69* | 94.15* | 92.59 | 93.96 | **94.78** |
| ResNet-8 | 88.14* | 87.26 | 88.71 | 88.37* | 88.38* | 89.54 | 90.12 | **91.63** |
| ResNet-32 | 93.72* | 93.71 | 93.75 | 93.73* | 93.24* | 92.20 | 93.80 | **94.11** |
| ResNet-50 | 95.81* | 95.92 | 95.70 | 95.44 | 95.79 | 91.90 | 96.18 | **97.24** |
| SENet-32 | 93.73 | 94.86 | 94.82 | 95.09 | 95.04 | 94.71 | 95.48 | **95.80** |
| MNetv2 | 90.94 | 91.56 | 91.25 | 91.04 | 90.71 | 91.39 | 91.80 | **92.97** |
| ViT-Tiny | 82.37* | 81.43 | 80.52 | 83.12* | 82.61* | 80.68 | 81.16 | **86.42** |
| Swin-Tiny | **85.93*** | 85.61 | 85.06 | 84.76* | 85.71* | 85.01 | 84.98 | 85.56 |
| Cait-XXS | 84.28* | 83.82 | 82.81 | 84.14* | 84.46* | 83.22 | 82.66 | **88.13** |

Table 3: Top-1 accuracy (%) on the CIFAR-100 dataset for various activation functions across different network models

| Networks | ReLU | LReLU | PReLU | GELU | GCLU | Maxout | Dy-ReLU | ADReLU |
|---|---|---|---|---|---|---|---|---|
| VGG-16 | 73.21* | 73.92 | 70.50 | 72.47* | 74.13* | 74.41 | 75.10 | **76.84** |
| ResNet-8 | 60.73* | 60.77 | 62.57 | 60.61* | 60.82* | 63.61 | 63.66 | **64.98** |
| ResNet-32 | 71.91* | 71.16 | 70.65 | 71.87* | 71.75* | 69.66 | 72.25 | **72.51** |
| ResNet-50 | 80.79* | 78.25 | 79.95 | 79.93* | 81.85* | 72.17 | 80.74 | **82.16** |
| SENet-32 | 76.26 | 76.27 | 76.55 | 77.63 | 78.32 | 75.63 | 78.08 | **80.21** |
| MNetv2 | 67.92 | 70.48 | 68.71 | 69.07 | 66.70 | 67.08 | 67.81 | **71.09** |
| ViT-Tiny | 55.59* | 54.10 | 53.50 | 57.81* | 56.63* | 52.69 | 53.54 | **58.75** |
| Swin-Tiny | **59.54*** | 58.68 | 58.43 | 57.25* | 58.92* | 57.11 | 57.93 | 59.24 |
| Cait-XXS | 59.16* | 58.85 | 58.97 | 59.77* | 59.67* | 58.06 | 58.03 | **65.91** |

As shown in Table 2, ResNet-50 to 97.24%, a 1.06% improvement over Dy-ReLU, and raising MobileNetV2 to 92.97%, 1.17% higher than with Dy-ReLU. Most notably, ADReLU drives substantial advances in transformer-based models, elevating ViT-Tiny to 86.42% (+3.30% over GELU) and achieving a remarkable 88.13% with CaiT-XXS, which outperforms its best baseline by 3.67%. The results on the more complex, fine-grained CIFAR-100 in Table 3, enhance ResNet-50 to 82.16%, a 0.31% gain over GCLU, and enable SENet-32 to reach 80.21%, outperforming GCLU by 1.89%. ViT-Tiny with ADReLU attains 58.75% accuracy, a +0.94% improvement over GELU. The most dramatic improvement is observed with CaiT-XXS, where ADReLU achieves 65.91%, a massive +6.14% absolute gain over the best baseline (GELU at 59.77%).

The results demonstrate that the ADReLU consistently surpasses baseline activations on both datasets, underscoring its versatility in enhancing feature extraction across architectures and task complexities. The only outlier is Swin-Tiny, where ADReLU remains competitive but falls short of the lead, highlighting opportunities for refinement to better integrate with its shifted-window attention.

**ImageNet classification**. The ImageNet dataset serves as a critical large-scale benchmark for evaluating the scalability and efficiency of novel neural network components. This experiment tests

ADReLU's ability to enhance performance in complex, real-world visual recognition tasks across models of varying capacities and depths.

Table 4: Top-1 test accuracy (%) on the ImageNet dataset for ReLU, Dy-ReLU, and ADReLU across ResNet variants (ResNet-10/18/34/50) and MobileNetV2 scales (×0.35, ×0.5).

| Activation Layer | ReLU | DY-ReLU | ADReLU |
|---|---|---|---|
| ResNet-10 | 63.0* | 66.3* | **68.3** |
| ResNet-18 | 69.8* | 71.8* | **72.2** |
| ResNet34 | 73.3* | 74.4* | **75.6** |
| ResNet-50 | 76.2* | 77.4* | **79.2** |
| MNetV2×0.5 | 65.4* | 70.3* | **72.7** |
| MNetV2 ×0.35 | 60.3* | 66.4* | **69.3** |

As shown in Table 4, on MobileNetV2 with scale value ×0.35, ADReLU achieved 69.3% accuracy, representing a remarkable +9.0% over the ReLU baseline (60.3%) and a +2.9% improvement over Dy-ReLU (66.4%). For MobileNetV2×0.5, ADReLU reaches 72.7%, outperforming ReLU and Dy-ReLU by +7.3% and +2.4%, respectively. The performance benefits extend consistently to deeper ResNet architectures. ADReLU attains 79.2% accuracy on ResNet-50, a +2.8% gain over the ReLU baseline. This trend is maintained across the entire ResNet family: ResNet-10 achieves 68.3% (+5.3% over ReLU), ResNet-18 reaches 72.2% (+2.4%), and ResNet-34 scores 75.6% (+2.3%).

ADReLU demonstrates exceptional scalability and performance on the large-scale ImageNet benchmark, confirming its practical utility for real-world applications. The most dramatic improvements on capacity-constrained models like MobileNetV2 underscore that its adaptive thresholds more effectively enhance feature learning and gradient flow.

***ADReLU evaluations on Pre-trained SOTA models***

The fine-tuning procedure integrates ADReLU into the pre-trained EfficientNetV2 family (Small, Medium, and Large), following the same parameter settings as Tan & Le (2021). We replace the last three SiLU activation layers of EfficientNetV2 with ADReLU, allowing its attention-driven thresholds to adaptively enhance high-level feature representations during fine-tuning. The results demonstrate the top-1 accuracy on CIFAR-100: the Small variant's accuracy improved slightly from 89.68% to 89.72%, the Medium from 90.87% to 91.02%, and the Large from 91.68% to 91.88%. These gains, though modest, show that even minimal substitution in the deepest layers of EfficientNetV2 can provide measurable benefits. Moreover, ADReLU serves as a drop-in replacement that boosts performance without requiring architectural modifications.

### 4.3 PARAMETER AND TIME COMPLEXITY ANALYSIS

To assess the practical deployment cost of ADReLU, we evaluate its parameter overhead and computational efficiency against key baseline activation functions: the static ReLU, the input-independent trainable PReLU, and the input-dependent Dy-ReLU.

Table 5: Parameter counts and total training/testing times (mm:ss) on the CIFAR-100 across ResNet variants with different activations, measured on an NVIDIA RTX 2080 GPU.

| Networks | ResNet-8 | | ResNet-32 | | ResNet-50 | |
|---|---|---|---|---|---|---|
| | #Params | Tr/Ts Time | #Params | Tr/Ts Time | #Params | Tr/Ts Time |
| ReLU | 83,892 | 25:06 / 03:14 | 472,756 | 65:53 / 06:40 | 23,705,252 | 155:32 / 11:40 |
| PReLU | 83,899 | 31:14 / 03:48 | 472,787 | 81:32 / 05:58 | 23,705,269 | 219:36 / 17:20 |
| Dy-ReLU | 98,672 | 56:46 / 04:13 | 545,104 | 147:08 / 17:55 | 52,100,052 | 358:06 / 27:20 |
| ADReLU | **96,604** | **42:54 / 03:40** | **533,148** | **129:12 / 13:20** | **30,234,140** | **331:40 / 23:30** |

The results in Table 5 show that integrating ADReLU into the ResNet-50 model reduces the total number of parameters by 42% compared to Dy-ReLU, highlighting its efficiency. The ADReLU model also trains in 331 minutes, a 7% faster time than the Dy-ReLU model at 358 minutes, and improves inference speed by 14% per epoch. For the smaller ResNet-8, ADReLU not only has fewer

parameters but also trains 24% faster than its Dy-ReLU counterpart. As anticipated, the activation functions like ReLU and PReLU have the fewest parameters and the fastest training times, thanks to their simple design. In contrast, while the dynamic nature of Dy-ReLU and ADReLU adds complexity, the ADReLU implementation improves network expressivity at a lower computational cost, making it suitable for resource-limited applications.

### 4.4 ANALYSIS

We examine the mechanisms of ADReLU through targeted analyses, including a case study of its performance in lightweight fully connected networks using CIFAR-10 and CIFAR-100 datasets. This study shows ADReLU's versatility beyond convolutional architectures. We also analyze sparsity patterns to assess activation efficiency in ResNet-8, ResNet-32, and ResNet-50 with CIFAR-10 validation samples. Additionally, we investigate the computational complexity and number of parameters of ADReLU compared to baseline activation functions like ReLU and Dy-ReLU across various ResNet architectures. These analyses highlight how ADReLU balances expressivity and sparsity-induced regularization, offering insights into its adaptability and advantages in diverse neural network applications.

***Case study: Validation of ADReLU in Simple Fully Connected Networks***

To demonstrate that ADReLU's benefits are not limited to convolutional models, we conducted a case study using a three-layer fully connected (FC) neural network on the CIFAR-10 and CIFAR-100 datasets, because using the simpler network means fewer factors will affect the accuracy, and it's more clear to exhibit the improvements caused by our ADReLU. The experiment utilizes a three-layer FC network, with each layer followed by an activation function. The network architecture consists of an input layer that flattens CIFAR images ($32 \times 32 \times 3$ ) into a dimensional vector, followed by linear layers transforming the dimensions as follows: $3072 \rightarrow 512$, $512 \rightarrow 256$, and $256 \rightarrow 128$, each followed by an activation ADReLU.

Table 6: Accuracy (Top-1) performance of a simple 3-layer FC network on the CIFAR-10 and CIFAR-100 datasets.

|  | CIFAR-10 | CIFAR-100 |
|---|---|---|
| ReLU | 62.79 | 34.41 |
| PReLU | 63.19 | 34.98 |
| DyReLU | 61.81 | 31.02 |
| AD-ReLU | **63.44** | **35.11** |

However, Table 6 presents the Top-1 accuracy (%) of the FC network on CIFAR-10 and CIFAR-100, with rows corresponding to activation functions and columns representing the datasets. ADReLU achieves 63.44% on CIFAR-10, slightly improving over ReLU by 0.65% and PReLU by 0.25%. On CIFAR-100, it achieves 35.11%, outperforming ReLU by 0.70% and PReLU by 0.13%. Although these gains are small in absolute terms, they underscore ADReLU's capacity to introduce adaptive, spatially aware activations that enhance feature representation and generalization in non-convolutional settings.

***Sparsity VS. Expressivity***

To investigate the internal behavior of ADReLU, we analyze the trade-off between expressivity and sparsity in selected models, ResNet-8, ResNet-32, and ResNet-50, by conducting the comparison experiments on CIFAR-10/100 datasets against ReLU, PReLU, and Dy-ReLU.

On the one hand, sparsity refers to the proportion of zero-valued output activations following a non-linear transformation. Here, the sparsity with $N$ neurons is defined as Eq. (7),

$$S = \mathbb{E}_{x \sim D}[\frac{1}{N} \sum_{i=1}^{N} \mathbf{1}(a_i(x) = 0)], \tag{7}$$

where $x \sim D$ denotes an input sampled from the dataset, $a_i(x)$ is the activation of unit $i$ for input $x$, and $1(\cdot)$ is an indicator function that equals 1 if the activation is zero, and 0 otherwise. It measures the fraction of neurons that remain inactive during a forward pass, reflecting the network's capacity

to suppress less informative features while emphasizing the most relevant ones. Sometimes, the sparsity is considered to be related to the robustness of the model.

On the other hand, expressivity is the capability of a model to express complex functions. When training is sufficient, the training accuracy reflects the expressivity of the model.

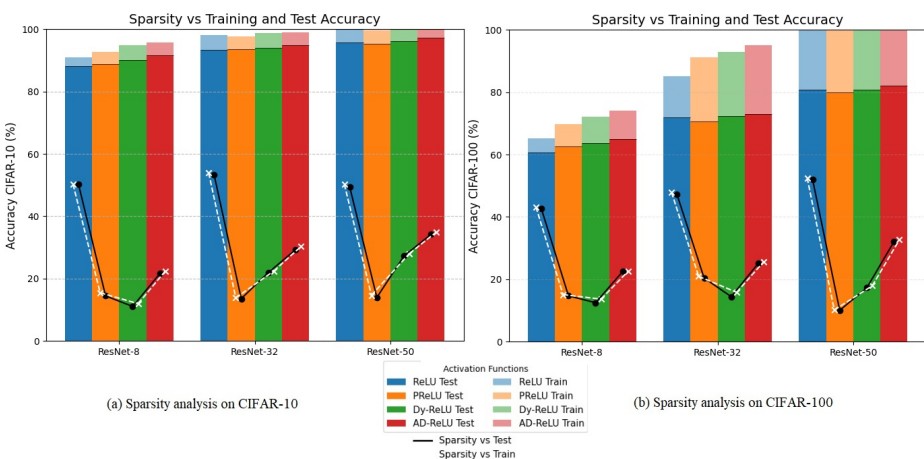

Figure 2: Sparsity vs. training/test accuracy across models and activations on CIFAR-10 and CIFAR-100

Figure 2 illustrates the sparsity (black/white line) alongside training and test accuracies. For the ResNet-8 model, ReLU's high sparsity suppresses excessive features, leading to the lowest accuracies and a small train–test gap. This gap, however, is not due to strong generalization but rather reduced expressivity caused by excessive feature suppression. In contrast, low-sparsity like PReLU and Dy-ReLU retain more features, yielding slightly higher expressivity and accuracy but also slightly larger gaps. ADReLU achieves a moderate sparsity, balancing suppression and retention. As a result, it improves test accuracy over baselines while maintaining a small train–test gap. As model depth increases to ResNet-32, ReLU's high sparsity introduces some regularization but still limits expressivity, resulting in suboptimal accuracy. PReLU and Dy-ReLU, with their low sparsity, tend to overfit by retaining excessive noise, thereby widening the train–test gap. ADReLU, with its input-dependent threshold, dynamically adapts sparsity and achieves higher expressivity while keeping the gap smaller than that of ReLU and Dy-ReLU. For deep models like ResNet-50, ReLU produces high sparsity by suppressing a large portion of activations. PReLU and Dy-ReLU yield lower sparsity, retaining more activations. And ADReLU adapts to a moderate sparsity, striking a balance between feature suppression. While training accuracies for all methods approach saturation—indicating strong expressivity—ADReLU shows a smaller gap between training and test accuracy compared to ReLU, PReLU, and Dy-ReLU. This reduced gap reflects better generalization, highlighting ADReLU's ability to maintain high expressivity while leveraging moderate sparsity.

## 5 CONCLUSION

In this work, we introduce ADReLU, a novel activation function that reexamines ReLU by incorporating attention mechanisms. Our key idea is that the activation threshold should be dynamic and input-dependent, adapting to local contexts and feature interactions. By using a computationally efficient QKV attention mechanism, ADReLU allows networks to make precise activation decisions, enhancing feature representation. Extensive experiments show that this effective approach consistently outperforms existing activation functions across various datasets and architectures. Additionally, our design choices—grouped and depth-wise projections—ensure that the increased expressivity does not lead to high computational costs, making ADReLU a practical drop-in replacement for standard activations.

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
