# OpenReview forum: "ADReLU: Enhancing Neural Network Expressivity with Attention-based Dynamic ReLU"
_ICLR.cc/2026/Conference — ICLR 2026 Conference Withdrawn Submission_

### Official Review · Reviewer_pK1A · 2025-10-25

**Soundness:** 2
**Presentation:** 2
**Contribution:** 2
**Rating:** 2
**Confidence:** 4

**Summary:**

This paper proposes a novel activation function named ADReLU, which dynamically determines the activation threshold for each neuron through an attention mechanism, replacing the fixed zero threshold in traditional ReLU. To maintain computational efficiency, the authors employ grouped convolutions and depthwise separable projections. Extensive experiments on multiple image classification benchmarks (CIFAR-10/100, SVHN, and ImageNet) and various architectures (ResNet, ViT, MobileNet) demonstrate that ADReLU consistently outperforms existing predefined and learnable activation functions in terms of accuracy.

**Strengths:**

# **Strengths**

1. **Originality:** The paper introduces the attention mechanism into the design of activation functions, proposing a dynamic and input-dependent thresholding mechanism, which demonstrates a notable level of innovation.

2. **Practicality:** By employing grouped convolutions and depthwise separable projections, the method effectively controls computational complexity, maintaining practical deployability while enhancing representational capacity.

3. **Comprehensive Experiments:** The experimental validation is thorough, covering multiple datasets and network architectures, including both CNNs and Transformer-based models. The results consistently show that ADReLU outperforms existing methods.

4. **Sound Analysis:** Beyond accuracy comparisons, the paper also investigates the relationships among sparsity, representational power, and generalization, which strengthens the credibility of the conclusions.

5. **Clarity of Presentation:** The paper is well-written and clearly structured, with detailed figures and mathematical formulations that make the methodology easy to follow and understand.

**Weaknesses:**

# **Weaknesses**

1. **Limited Novelty:** The core idea of ADReLU is similar to Dy-ReLU, with the main difference being the addition of an attention mechanism. However, the paper does not provide a detailed explanation of why this particular attention design is superior to other possible adaptive mechanisms, making the novelty less prominent.

2. **Insufficient Theoretical Analysis:** Although the paper provides intuitive explanations and sparsity analysis, it lacks formal theoretical derivation or convergence analysis. For example, the distribution characteristics of the dynamic threshold and its impact on gradient propagation could be quantified further.

3. **Incomplete Baseline Comparisons:** Existing experiments only compare ReLU, LReLU, PReLU, GELU, GCLU, Maxout, and Dy-ReLU. Recent mainstream activation functions such as AReLU, Swish, SiLU, HardSwish, GeGLU, and SwiGLU are not included.

4. **Insufficient Ablation Studies:** The contributions of individual components in ADReLU (e.g., number of groups, attention subspace dimensions, normalization methods) are not systematically investigated, limiting understanding of the design choices.

5. **Unexplained Swin-Tiny Results:** ADReLU does not achieve optimal performance on Swin-Tiny, and the authors merely mention “to be further optimized” without analyzing compatibility issues with the windowed attention mechanism. This lack of analysis may affect readers’ assessment of ADReLU’s adaptability and hinder future improvements.

6. **Limited Generalization Evaluation:** All experiments are focused on classification tasks, with no evaluation on downstream tasks such as detection or segmentation, which limits the convincingness of its applicability to broader vision problems.

**Questions:**

# **Questions**

1. **Baseline Selection:** The authors did not include recent mainstream activation functions such as Swish or SwiGLU in the comparisons. Is this due to technical limitations (e.g., insufficient computational resources) or theoretical considerations (e.g., considering these functions as belonging to a different design paradigm than ADReLU)? It is recommended to supplement the study with comparisons to these baselines.

2. **Theoretical Analysis:** Could the authors provide theoretical or intuitive analysis of ADReLU's enhanced representational capability, for example from the perspectives of gradient propagation or function approximation? This would strengthen the theoretical credibility of the conclusions.

3. **Stability of Dynamic Threshold τ:** Did the authors observe significant fluctuations in τ during early training or across different batch distributions? Are there any regularization or gradient constraints to stabilize this threshold? How does the dynamic threshold affect gradient flow stability?

4. **Hyperparameter Selection:** What is the rationale for choosing g=4 and dk=8 as hyperparameters in ADReLU? It is suggested to provide supporting experiments or ablation studies.

5. **Abnormal Results:** Regarding the underperformance of Swin-Tiny on CIFAR-10 and CIFAR-100, are there any preliminary investigations or explanations?

6. **Extension to Other Tasks:** ADReLU has been mainly evaluated on image classification tasks. Have the authors considered applying it to other computer vision tasks (e.g., object detection, semantic segmentation)? Does it show similar performance improvements in these scenarios?

7. **Scalability to Larger Models:** The current experiments focus on small to medium-scale models. Are there results for larger-scale models (e.g., EfficientNet-L, ViT-L, Swin-B)? Is there any saturation effect observed when scaling up?

---

### Official Review · Reviewer_MSjC · 2025-10-30

**Soundness:** 3
**Presentation:** 2
**Contribution:** 3
**Rating:** 4
**Confidence:** 5

**Summary:**

This study introduces an innovative activation approach derived from the concept of ReLU. Instead of relying on a constant zero cutoff, the proposed method employs an adaptive, input-sensitive threshold determined through an attention-based process. This design allows the activation behavior to adjust dynamically according to the characteristics of the input data.

**Strengths:**

A different approach involves dynamically adjusting the model according to the characteristics of the input data using an attention mechanism. Incorporating attention mechanisms into the activation function introduces a unique idea, combining the both convolutional operations and attention. When the activation function is applied, the entire input can be converted into a single threshold for values where $x \leq 0$, enhancing the model’s efficiency.

**Weaknesses:**

See Question Section Below.

**Questions:**

1) In Figure~1, within the attention mechanism, why is the group size fixed according to the number of channels in the $1\times1$ convolution ($g$)?

2) In the theoretical analysis presented in Section~3, the mechanism of ADReLU provides information only about the forward propagation, specifically describing the behavior for $x \leq 0$. However, the explanation regarding the backpropagation dynamics is not provided. Could the authors clarify how ADReLU affects the backward pass?

3) From lines~167--173, the time complexity comparison is presented between the traditional convolution and the ADReLU activation function alone. However, when ADReLU is used, the total time complexity becomes $\mathcal{O}(C^2HW) + \mathcal{O}(CHW) \approx \mathcal{O}(C^2HW)$. Therefore, comparing the convolution and activation function complexities separately may not provide a fair assessment. Could the authors justify this comparison approach?

4) From line~181, the paper claims that ADReLU is suitable for Vision Transformer (ViT) architectures. However, on the ImageNet dataset, ViT experiments is performed , and the evaluation appears to be limited to ReLU-based activation functions. To strengthen the claim, it would be more appropriate to include experiments using transformer-based architectures, comparing ADReLU with GELU, which is commonly adopted in such models.

5)  As per in Figure 1, the final "A'' matrix appears to have a spatial dimension of $1 \times 1$. However, in line 140, the "A'' matrix is multiplied by $V$, which would require it to have the same spatial dimensions as the input $X$. This suggests that a connection or operation may be missing in Figure~1. Moreover, the equations from lines 150 to 155 do not align with the structure depicted in the diagram, and there is no clear explanation of the dynamic threshold or the necessity of the associated operations.

6) In the attention mechanism, we observed the following: (i) When images are **not normalized** before being passed into the architecture, an image containing approximately 15% of black/grey pixels (pixel values in the range 0-40) results in the sigmoid output being close to 1. (ii) When images are **normalized** before being passed into the architecture, even 1% of image can give value from sigmoid function directly equal to 1.
In such cases, the (Q) and (K) computations may become unnecessary, as the sigmoid already saturates to 1, meaning only the (V) component effectively contributes to the dynamic threshold.

        Question: For all experiments in this paper, were the images normalized or not before being passed into the architecture?

7) We suggest that the authors include performance graphs to illustrate the comparative results and trends, which would enhance the understanding of the findings.

---

### Official Review · Reviewer_CshK · 2025-10-31

**Soundness:** 2
**Presentation:** 2
**Contribution:** 2
**Rating:** 0
**Confidence:** 5

**Summary:**

This paper introduces ADReLU (Attention-based Dynamic ReLU), a activation function designed to enhance neural network expressivity by replacing ReLU’s fixed zero threshold with a dynamic, input-dependent threshold computed via an attention mechanism. ADReLU uses a lightweight QKV-style attention module with grouped and depthwise convolutions to balance flexibility and efficiency. Extensive experiments on CIFAR-10/100, SVHN, and ImageNet across CNNs and Vision Transformers show consistent accuracy improvements over both static (ReLU, LReLU) and trainable (PReLU, GELU, Dy-ReLU) activations. The authors also analyze sparsity, complexity, and generalization, demonstrating that ADReLU improves representation power while maintaining computational efficiency.

**Strengths:**

ADReLU demonstrates promising performance on small-scale datasets, and also shows noticeable improvements on ImageNet when applied to relatively simple architectures such as ResNet.

**Weaknesses:**

* As an activation function, ADReLU introduces a substantial number of learnable parameters and additional computational overhead, which seems inappropriate for its stated role. Rather than a lightweight activation design, ADReLU functions more like an independent network module. Consequently, its superior results on small-scale datasets may not be entirely convincing, as such datasets are particularly sensitive to model capacity and parameter size. The observed gains might therefore stem from the increased parameterization and computation rather than from the structural innovation itself.

* Attention-based activation functions have been explored in earlier works, such as **AReLU** ([arXiv:2006.13858](https://arxiv.org/abs/2006.13858)), which demonstrated notable advantages without introducing many new parameters or complex computations—sometimes with only a few trainable hyperparameters. The authors should include comparisons or discussions with this line of work to provide a more complete analysis.

* In the current landscape dominated by Transformer-based architectures, the potential performance gain from incorporating ADReLU may be rather limited.

**Questions:**

Regarding the sparsity analysis in Figure 2, I find the authors’ interpretation somewhat unconvincing (or perhaps I have misunderstood it, and I hope the authors can clarify this point in the rebuttal). To my knowledge, there is currently no established evidence showing a direct causal relationship between the degree of activation sparsity and overall model performance. On the contrary, one could argue that fewer activations might indicate that the extracted features are more representative and discriminative. From other perspectives—such as model acceleration, pruning, or quantization—higher sparsity (i.e., fewer active features) can actually be beneficial for model efficiency. Therefore, I do not necessarily agree that the lower activation rate produced by ReLU should be viewed as a disadvantage.

---

### Official Review · Reviewer_ZpxZ · 2025-10-31

**Soundness:** 2
**Presentation:** 3
**Contribution:** 2
**Rating:** 2
**Confidence:** 4

**Summary:**

- The paper introduces ADReLU, an activation function that replaces ReLU’s fixed zero threshold with an input-dependent threshold computed via an attention mechanism.
   - To balance expressivity and computational efficiency, the authors incorporate grouped convolution and depth-wise projection.
- Extensive experiments on CIFAR-10/100, SVHN, and ImageNet demonstrate consistent performance improvements across CNNs and Transformers, e.g., a +2.8% accuracy gain over ReLU on ResNet-50 (79.2% top-1).
- The authors also analyze sparsity patterns to reveal a balance between expressivity and generalization.

**Strengths:**

- ADReLU shows substantial and stable improvements over both predefined and dynamic activations across diverse datasets and architectures.

- The combination of attention-driven dynamic thresholds and grouped/depth-wise operations effectively enhances adaptability without heavy computational overhead.

- Experiments include CNNs, Transformers, and even a simple fully connected network, along with sparsity–expressivity analysis that supports the method’s general effectiveness.

- ADReLU can be seamlessly integrated as a drop-in replacement for ReLU, requiring no architectural modification.

**Weaknesses:**

- The experiments are confined to image classification, lacking evaluations on detection, segmentation, or non-visual tasks.

- The attention-based design relies heavily on convolutional feature maps, making it less generalizable to non-visual domains.

- Lack of theoretical insight. While empirical analyses are solid, the underlying reason why attention-based thresholding improves learning remains insufficiently explained.

-The gain on Swin-Tiny and some transformer models is minor, suggesting potential compatibility issues with windowed attention mechanisms.

**Questions:**

- Can ADReLU be extended or adapted to non-visual or sequential tasks, such as NLP or time-series modeling?

- Could you provide more interpretability, e.g., visualizations or ablations, about how the attention-generated thresholds affect feature activation?

- Why is the performance gain smaller on Swin-Tiny? Is it related to its shifted-window design?

- How does ADReLU affect training stability or convergence dynamics compared to ReLU and Dy-ReLU?

- Have you evaluated its applicability to resource-constrained environments (e.g., TinyML or mobile devices)?

---

### Note · Authors · 2025-11-14

I have read and agree with the venue's withdrawal policy on behalf of myself and my co-authors.